# Multiscale Modeling and Simulation of Polymer Blends in Injection Molding: A Review

**DOI:** 10.3390/polym13213783

**Published:** 2021-10-31

**Authors:** Lin Deng, Suo Fan, Yun Zhang, Zhigao Huang, Huamin Zhou, Shaofei Jiang, Jiquan Li

**Affiliations:** 1School of Mechanical and Electrical Engineering, Wuhan Institute of Technology, Wuhan 430205, China; dengl@wit.edu.cn (L.D.); fan_suo@wit.edu.cn (S.F.); 2State Key Laboratory of Material Processing and Die & Mold Technology, Huazhong University of Science and Technology, Wuhan 430074, China; marblezy@hust.edu.cn (Y.Z.); hmzhou@hust.edu.cn (H.Z.); 3College of Mechanical Engineering, Zhejiang University of Technology, Hangzhou 310014, China; jsf75@zjut.edu.cn (S.J.); Lijq@zjut.edu.cn (J.L.)

**Keywords:** polymer blends, injection molding, microstructure, multiscale simulation

## Abstract

Modeling and simulation of the morphology evolution of immiscible polymer blends during injection molding is crucial for predicting and tailoring the products’ performance. This paper reviews the state-of-the-art progress in the multiscale modeling and simulation of injection molding of polymer blends. Technological development of the injection molding simulation on a macroscale was surveyed in detail. The aspects of various models for morphology evolution on a mesoscale during injection molding were discussed. The current scale-bridging strategies between macroscopic mold-filling flow and mesoscopic morphology evolution, as well as the pros and cons of the solutions, were analyzed and compared. Finally, a comprehensive summary of the above models is presented, along with the outlook for future research in this field.

## 1. Introduction

The product performance of immiscible polymer blends is significantly affected by their microstructure, which is formed during mixing and processing. Most polymer blend products are made by injection molding. Therefore, an accurate description of the morphology evolution in polymer blends during the injection molding process is the necessary condition for predicting and tailoring their final properties [1]. However, the simulation of the injection molding of polymer blends is a challenging task. The mold-filling flow of polymer blend melt spans macro- and mesoscales, and no single model is capable of simulating these complex processes on both scales at the same time [2]. Therefore, a multiscale modeling and simulation approach is necessary for the injection molding process of polymer blends. In this paper, recent advances in the development of macroscopic and mesoscopic models are reviewed for the injection molding of polymer blends as well as scale-bridging methodologies. Here, this review restricts its attention to papers that connect microstructure with flow and rheology; the flow field is well defined. A review of all the publications pertaining to polymer blends with surfactants or compatibilizers would go beyond our scope limitations, though they are technologically important.

The rest of the paper is organized as follows. In Section 2, the process of mold-filling of polymer blends is described on both macro- and mesoscales, and the framework of multiscale modeling is thoroughly described. In Section 3, the historical development of the mold-filling flow simulation of polymer melt is overviewed. In Section 4, mesoscopic models for droplet morphology evolution during processing are compared. Different scale-bridging methodologies between microstructure evolution and mold-filling flow are discussed in Section 5. In Section 6, a summary is made, and an outlook for the future is suggested.

## 2. Framework of Multiscale Modeling

Multiscale modeling and simulation of polymer processing has been a hot research area, but it should be recognized that it is not a burgeoning field formed only during the past decade. Its development goes back to the computational fluid dynamics of polymer melt flow and the exploration of the molecular structure of polymers, both of which have laid the foundation of multiscale research. Therefore, the multiscale modeling and simulation of polymer processing ought to cover the consensus of available research, i.e., increasingly sophisticated injection molding simulation, and focus on the spawned problems when coupled with different models.

As shown in Figure 1, polymeric materials have a unique hierarchical structure, from atom to monomer, chain, and conformation with corresponding simulation methods on each scale. For the injection molding of polymers, the orientation, stretching, and crystallization of the polymer molecules occurs on the microscale, while on the macroscale, the polymer melt fills the mold cavity as continuum fluid.

For polymer blends, it is mesoscopic morphology that has the most direct influence on the mold-filling flow of polymer blend melt and the final performance of the products. The interest in the multiscale modeling and simulation of injection molding of polymer blends is focused primarily on the macroscopic mold-filing of polymer blend melt and the simultaneous mesoscopic morphology evolution.

**Macroscale (~10^−3^ m, ~1 s):** The solid plastic pellets melt at a certain temperature above the melting point and are injected into the mold at a certain speed under the pressure of the injection machine. The polymer melt expels the air from the cavity until it fills the entire mold cavity and finally cools and solidifies to get the product as the designed mold cavity. The polymer blend melt is considered to be a continuous fluid, with the inside microstructure neglected. Injection molding of polymer melt is a non-Newtonian, non-isothermal, and unsteady process of mass, momentum, and heat transfer with the moving polymer–air front.

**Mesoscale (~10^−6^ m, ~10^−6^–10^−3^ s):** Mesoscale morphology is formed in the compounding and granulation stages prior to injection molding. In the equipment, such as roll mills, mixing machines or screw extruders, a sea-island-like multiphase structure emerges [3,4] when the micron-sized droplets are dispersed throughout the matrix. During the injection molding stage, the dispersed droplets inside the blend melt undergo complex morphology evolution under the combined action of shear, pressure and heat, and interfacial tension [5]. Finally, the blend morphology is frozen inside the product as the polymers cool and solidify after the flow ceases.

The macroscopic and mesoscopic physics of the injection molding of polymer blends are described using separate models. The mold-filling flow of the polymer melt is a free surface flow and can be described using the fields of velocity, pressure, temperature, and phase fraction. Using an appropriate constitutive equation and a PVT state equation of polymer melt, the conservation equations for the mold filling flow can be closed, and, thus, the mold filling flow can be modeled on a macroscopic scale. Using different discretization methods, such as the finite difference (FD), finite element (FE), or finite volume (FV) methods, the above equations can be solved numerically and the mold-filling flow process of the polymer melt can be accurately simulated.

In contrast, there is no unified approach for the description of the polymer blends morphology, and, accordingly, there are different modeling approaches for the morphology evolution of polymer blends. One type of method makes certain assumptions about and parameterizes the shape of the dispersed phase droplets and then establishes the evolution equations for the morphological parameters of the polymer blends. This class of methods is relatively simple, easy to implement, and computationally low but is only applicable under limited conditions. A different class of models uses the polymer molecular structure, coarse-grained particles, and component concentrations to describe the polymer blend morphology, thus modeling the evolution of the morphology of the blend separately. These methods are free of the assumption of polymer blend morphology and are more applicable to various cases; however, the computational overhead is usually expensive.

Given the macroscopic models of the mold-filling flow of polymer melt and the mesoscopic models of the polymer blend morphology evolution established, the key to the success of multiscale modeling and the simulation of injection molding of polymer blends is revealing the interactions between the physics of different scales and developing corresponding scale-bridging strategies. As discussed above, the multiple fields of velocity and pressure, as well as the temperature of the mold-filling flow, are the driving force of the morphology evolution of polymer blends. In turn, the morphology makes the stress–strain response characteristics of the polymer blend markedly different from a simple linear combination of the components. Therefore, various scale-coupling methods are used in different ways to represent the factors of mesoscopic morphology in the macroscopic model of the mold-filling flow. 

According to the analysis of injection molding on macro- and mesoscales, the modeling and simulation of injection molding of polymer blends can be decomposed into three ingredients: macroscopic mold-filling flow modeling, mathematical characterization of polymer blend morphology and the modeling of its evolution, and, more importantly, the coupling of the models on macro- and mesoscales. This paper is dedicated to reviewing the multiscale modeling and simulation of injection molding of polymer blends from three aspects: macroscopic mold-filling flow simulation, mesoscopic morphology evolution simulation, and scale-bridging methods, corresponding to the following sections.

## 3. Mesoscopic Modeling of Droplet Morphology Evolution

Before articulating the mesoscopic modeling methods of droplet morphology evolution, it is better to revisit the typical experiment of droplet morphology evolution in a typical flow field of injection molding. From the SEM observation of Figure 2, it can be seen that in the initial stationary state, the droplets appear as spheres with a relatively large volume mean radius and radius distribution range. When the shearing flow starts, the droplets are subjected to strong shear, and the droplet morphology gradually changes from spherical to ellipsoidal, and then, finally, to elongated fibrous, at a certain angle to the shear direction, while some droplets break up and produce two or more sub-droplets. When the shear flow field is weakened and stopped, the droplet deformation degree decreases and the shape starts to return to a spherical shape with a smaller average radius and a more uniform-sized distribution than in the initial state.

Many researchers have contributed to the numerical and experimental work of the morphology evolution of single droplets. In general, both the phenomenological and theoretical models can be roughly divided into three categories: droplet models based on ellipsoid approximation, phase field methods based on the Cahn-Hilliard equation, and the pseudo-potential model based on the lattice Boltzmann method.

### 3.1. Ellipsoid Droplet Models

Doi and Ohta proposed a theoretical model using the interfacial area per unit volume *Q* and interface tensor ***q*** to characterize the microstructure.
(1)$$Q\equiv \frac{1}{V}\int_{A}dA\;\mathrm{and}\;q=\frac{1}{V}\int_{A}\left(nn-\frac{1}{3}I\right)dA$$
where *A* represents the interfacial surface contained in *V*, with each surface increment *dA* having a unit normal vector ***n***, and ***I*** is the unit tensor.

They wrote the time derivative of each variable as the sum of two terms: one for external flow (deformation) and one for relaxation due to interfacial tension: (2)$$\dot{Q}={\dot{Q}}_{flow}+{\dot{Q}}_{relax}\;\mathrm{and}\;\dot{q}={\dot{q}}_{flow}+{\dot{q}}_{relax}$$

The Doi and Ohta model is applicable to arbitrary polymer blend morphology; however, due to the underlying closure problem, different expressions for the convective terms have been proposed, leading to convective nonlinearities.

For a droplet in simple shear field, when the viscosity ratio *p* is much larger than 1 and the volume fraction is below 1, the droplet will maintain an ellipsoidal shape for a wide range of capillary numbers and viscosity ratios. When the critical breakup scenario is approached, the droplet shape will deviate from the ellipsoidal assumption and the model itself will fail.

#### 3.1.1. Basic Quantities

As shown in Figure 3 below, three quantities are usually defined to characterize a deformed droplet in a planar flow field: the equivalent radius *R*, the deformation degree *Df*, and the orientation angle *θ*, where *Df* = (*a* − *b*)/(*a* + *b*), and *a* and *b* are the length and width of the droplet in the *x* − *y* plane of the flow, respectively. *R*, *a* and *b* have the relation of *R*^3^ = *ab*^2^. It is apparent that a larger *Df* indicates a more deformed droplet. *θ* is the angle between the principal axis of the droplet and the reference direction, usually the flow direction surrounding the droplet.

The study of the fundamentals of droplet deformation began with Taylor [7], who studied the basics of the deformation and breakup of a single Newtonian droplet in a viscous matrix. It was found that the deformation and breakup of the sheared droplet depended on two dimensionless quantities, namely, the droplet–matrix viscosity ration *p* = *η_d_*/*η_m_* and the capillary number *Ca* = *η_m_**γR*/Γ, where *η_d_* and *η_m_* are the viscosities of the droplet and the matrix, *γ* is the shear rate, and Γ is the interfacial tension between the phases of the droplet and matrix. The capillary number characterizes the competition of the viscous stress that drives the droplet deformation against the interfacial tension that maintains the original shape. Droplets exhibit different orientations when different forces dominate, as described in detail below.

#### 3.1.2. Deformation

Based on the ellipsoid description, the droplet shape can be described by a second-order tensor *G*, whose eigenvalues are equal to the reciprocal of the ellipsoidal semi-axis length squared, and the corresponding eigenvector defines the orientation angle of the droplet. The evolution of ***G*** can be expressed by the following equation [8]:(3)$$\frac{DG_{ij}}{Dt}+L_{ki}G_{kj}+G_{ik}L_{kj}=0$$
where $\frac{D}{Dt}$ represents the material derivative, *L* is the velocity gradient tensor inside the droplet, and *i* and *j* denote the space index and can be *x*, *y,* and *z*.

Once the velocity gradient tensor *L* is known, Equation (3) can be solved, and then the shape evolution of the ellipsoidal droplet will be uniquely determined with a proper initial condition. In fact, the most significant difference among the ellipsoid droplet models lies in how to calculate the velocity gradient tensor *L*.

Maffettone-Minale (MM) model [9]: Velocity gradient tensor *L* is expressed as:(4)$$L_{ij}=w_{ij}^{A}+f_{2}e_{ij}^{A}+\frac{f_{1}}{2\tau_{r}}\left(\frac{3G_{ij}}{G_{kk}}-\delta_{ij}\right)$$
where *τ**_r_* = η_m_*R*/Γ is the surface-tension relaxation time, *e_ij_* and *w_ij_* are the deformation rate tensor and the vorticity tensor, respectively, both of which can be obtained from the velocity gradient tensor *L_ij_*: *e_ij_* = (*L_ij_* + *L_ji_*)/2, *w_ij_* = (*L_ij_* − *L_ji_*)/2. *e_ij_* with superscript A denotes the externally applied deformation rate, while *e_ij_* without superscript is related to the deformation of the droplet itself. 

The parameters *f*_1_ and *f*_2_ are phenomenological model parameters for which certain choices have been proposed:(5)$$f_{1}=\frac{40(p+1)}{\left(2p+3)(19p+16\right)},f_{2}=\frac{5}{2p+3}+\frac{3Ca^{2}}{2+6Ca^{2}}$$

The above model is referred to as the MM2 model. When *f*_2_ is simplified to $f_{2}=5/(2p+3)$ the model is referred to as the MM1 model.

Jackson-Tucker (JT) model [10]: In a coordinate system coinciding with the main axes of the droplet, the velocity gradient tensor *L* can be expressed as:(6)$$L^{\prime }=fL_{Eshelby}^{\prime }+(1-f)L_{slender}^{\prime },p<0.1$$
(7)$$L^{\prime }=L_{Eshelby}^{\prime },p\geq 0.1$$
where f is a composite parameter depending on the dimensionless length of the droplet’s semi-major axis and equivalent radius *a*/*R*. $L_{Eshelby}^{\prime }$ and $L_{slender}^{\prime }$ represent the velocity gradient components obtained from the solution of Eshelby’s theory [11] for elastic ellipsoidal particles in elastic media and Khakhar and Ottino’s proposed filamentary theory [12], respectively.

Yu model [13]. The velocity gradient tensor *L* was expressed as:(8)$$L_{ij}=w_{ij}^{A}+(B_{mnkl}+C_{mnkl})r_{im}r_{uk}e_{uv}^{A}r_{vl}r_{jn}+r_{im}(\bar{L_{mn}^{\alpha }}+L_{mn}^{\beta })r_{jn}$$
where *r_ij_* is the transformation matrix that rotates the droplet axis in alignment with the coordinate system.

The velocity gradient tensor *L* in the Yu model consists of two major components: the components due to the external flow and the other one due to interface dynamics. The former of these can be referred to as the corresponding part of the JT model, for example. The meanings of the symbols in the equations and the detailed solutions can be found in the corresponding literature [13].

In addition to the above theoretical models, several traditional empirical models have found a wide application in simulations for simplicity and ease of use [14], such as the affine deformation model and Cox’s theoretical formulation [15].

The affine deformation model [16] assumes that the lengths of the two minor axes, *B* and *W*, always keep equal when the droplet is deformed. The major axis *L*, minor axes *W* and *B*, and the angle between the major axis and flow direction *θ* are calculated as:(9)$$L/R=0.5\gamma +0.5\sqrt{4+\gamma^{2}}$$
(10)$$B/R={\left(0.5\gamma +0.5\sqrt{4+\gamma^{2}}\right)}^{-0.5}$$
(11)$$\tan\theta ={\left(0.5\gamma +0.5\sqrt{4+\gamma^{2}}\right)}^{-1}$$
where shear rate *γ* represents the flow field that the droplet experienced.

The Cox theoretical formulation [15] is applicable to systems in all ranges of viscosity ratios when interfacial tension and viscous forces act simultaneously, and the predictions of deformation degree and orientation angle are given by:(12)$$Df=Ca\frac{19p+16}{16p+16}\frac{1}{\sqrt{{\left(19pCa/20\right)}^{2}+1}}$$
(13)$$\theta =\frac{1}{4}\pi +\frac{1}{2}\tan^{-1}(\frac{19pCa}{20})$$

#### 3.1.3. Breakup

When the capillary number *Ca* is slightly larger than the critical value *Ca_crit_*, the pattern of droplet breakup depends on the viscosity ratio [2]: when p is much smaller than 1, the droplet is extremely stretched into an S-shape and small droplets are released at both ends; when *p* ≈ 1, the droplet gradually necks out from the middle part until it breaks up into two sub-droplets, with some smaller satellite droplets in between. When *Ca* far exceeds *Ca_crit_*, the droplet stretches into an elongated fiber.

Based on the results of experimental studies on the droplet deformation and breakup mechanism under simple flow conditions, a simplified capillary number *k** = *Ca*/*Ca_crit_* (i.e., the ratio of the local capillary number to the critical value) can be defined. Depending on the cases, most researchers [14,17,18] have adopted the following general rule to describe droplet behavior.
(1)*k** < 0.1, droplets do not deform;(2)0.1 < *k** < 1, droplets deform, but do not break up;(3)1 < *k** < 4, droplets deform and split into two major sub-droplets;(4)*k** > 4, droplets form fibers with the affine deformation of the medium.

*Ca_crit_* decides whether the deformed droplets reach equilibrium shapes or breakup into sub-droplets. *Ca_crit_* of a specific droplet depends on the viscosity ratio and the ambient flow field type. The following empirical de Bruijn formula [19] is commonly used to write *Ca_crit_* as a function of p for simple shear fields [20].
(14)$$\mathrm{lg}Ca_{crit}=-0.506-0.0995\mathrm{lg}p+0.124{\left(\mathrm{lg}p\right)}^{2}-\frac{0.115}{\mathrm{lg}p-\mathrm{lg}4.08}$$

The breakup of a droplet into two major sub-droplets by necking can be calculated using the direct method. Assuming that a droplet of equivalent diameter *D*_0_ splits completely into two sub-droplets of the same diameter without considering the effect of surrounding droplets, the diameter of the split sub-droplet, *d* ≈ 0.794*D*_0_, can be obtained according to the principle of volume conservation.

Based on the definition of the actual time required for breakup, *t_b_*, a statistically significant average of the breakup process, can also be calculated. The rate of change of the total number of droplets *N_d_* is first obtained as:(15)$$\frac{dN_{d}}{dt}=\frac{\dot{\gamma }N_{d}}{Ca_{crit}t_{b}^{*}}$$
where $N_{d}=\frac{6\phi V}{\pi D_{0}{}^{3}}$, *V* is the total volume of the solution and *D*_0_ is the droplet diameter. This leads to the rate equation for droplet breakup [18].
(16)$${\left(\frac{dD_{0}}{dt}\right)}_{break}=\frac{-\dot{\gamma }D_{0}}{3Ca_{crit}t_{b}^{*}}$$

By comparing the above two models, it is easy to find that the direct method is simple and fast to calculate but cannot reflect the change trend in the breakup process. The physical meaning of the statistical method is clearer and scalable and the rate of change formula is easy to couple with others; however, the calculation cost is relatively high.

There are limited studies on droplet models for filament breakup processes, and most of them use empirical formulas or fluid-dynamics-related theories for approximate calculations. For example, based on the Rayleigh theory of capillary number instability [21] and the principle of volume conservation, the following equation can be used to calculate the final size of the fiber after the breakup [14]:(17)$$R_{drops}=R_{0}\sqrt[{3}]{\frac{3\pi }{2X_{m}}}$$
where *R*_0_ is the fiber diameter before the breakup, which can be approximated as the critical diameter *d** at the moment of fiber breakup, and *X_m_* represents the main wave number. When *p* = 1, *X_m_* ≈ 0.56, the diameter of the droplet after breakup *d* ≈ 2*d** is obtained using the above equation.

#### 3.1.4. Coalescence

For the coalescence process of two spherical particles of the same size in a shear flow field, the coalescence rate can be expressed as a function of collision probability and the kinetics of the collision process. Collision probability *p_coll_* can be expressed as (see [14]):(18)$$p_{coll}=\exp\left(\frac{-\pi }{8\dot{\gamma }\phi t_{loc}}\right)$$
where *t_loc_* is the local residence time.

The liquid film discharge probability *p_exp_* depends on the activity of the interface, which depends on the viscosity ratio *p*: the larger the value of *p*, the less active the interface is. For inactive interfaces with *p* much larger than 1, the liquid film discharge probability can be calculated using the following equation [14]:(19)$$p_{exp}=\exp\left[-\frac{9}{8}{\left(\frac{R}{h_{c}}\right)}^{2}k^{*}{}^{2}\right]$$

In the above equation, *h_c_* is the critical thickness of the liquid film at the time of the breakup, which can be obtained by experiment, and *k** is defined as before.

Thus, the coalescence chance of droplets *p_coal_* can be expressed as the product of *p_coll_* and *p_exp_*.

Considering the role of various factors in the coalescence process, the following equation for the evolution of the diameter of the dispersed phase in the coalescence process can be obtained [18].
(20)$${\left(\frac{dD_{0}}{dt}\right)}_{coal}=CD_{0}^{-1}\phi^{8/3}\dot{\gamma }$$
where *C* is the coalescence constant.

Take the assumption that the shape change of droplets during breakup and coalescence can be linearly superimposed; the resultant droplet diameter change rate can be calculated from Equations (15) and (19) as [18].
(21)$$\left(\frac{dD_{0}}{dt}\right)={\left(\frac{dD_{0}}{dt}\right)}_{break}+{\left(\frac{dD_{0}}{dt}\right)}_{coal}$$

The above equation can be integrated in time by various numerical methods, such as the finite difference method, to obtain the change in droplet diameter under the combined effect of breakup and coalescence.

Since the rate of change of droplet size at equilibrium is 0, according to the above Equations (15), (19), and (20), then we have [18]:(22)$$D_{eq}=D_{eq}^{0}+{\left(6CCa_{crit}t_{b}^{*}\phi^{8/3}\right)}^{1/2}$$
where *D_eq_* is the diameter at equilibrium and $D_{eq}^{0}$ is the diameter at zero component (i.e., without any dispersed phase), obtained by extrapolation. Using the above equation, the coalescence constant *C* of a certain blend can be obtained by preparing a series of materials with different components, i.e., calculated by the slope of the curve of *D_eq_* versus *ϕ*^4/3^. Using the coalescence probability formula obtained earlier, an approximate formula for the radius of the new particle after coalescence can also be obtained. Assuming the interaction of two droplets of the same size, the volume conservation principle yields (see [22]):(23)$$R^{*}=R{\left(\frac{2}{2-p_{coal}}\right)}^{1/3}$$
where *R* and *R** represent the radii of droplets before and after coalescence. Obviously, *R* = *R** when coalescence is not considered, i.e., *p_coal_* = 0; *R* = 2^1/3^*R** when coalescence is complete, i.e., *p_coal_* = 1.

Although the above experimental and probabilistic models cannot track the interfacial changes during droplet coalescence in real-time, they can reflect the changes of dispersed phase size in each part of the system due to the coalescence process and are simple and easy to implement.

#### 3.1.5. Size Distribution

For a discrete model of a system (droplet volume is a product of an integer and the elementary volume *V*_1_), the following equation describes the change in the number of droplets of volume *kV*_1_, *n_k_* with time *t* [23]:(24)$$\begin{array}{c} \frac{dn_{k}}{dt}=\frac{1}{2}\sum_{i+j=k}C\left(i,j\right)n_{i}n_{j}-F\left(k\right)n_{k}-\sum_{j=1}C\left(k,j\right)n_{k}n_{j}\\ +\sum_{j=k+1}\omega \left(k,j\right)n_{f}\left(j\right)F\left(j\right)n_{j} \end{array}$$
where *C*(*i*, *j*) is the coagulation kernel, *F*(*i*) is the overall breakup frequency, *n_f_*(*i*) is the number of fragments formed at the breakup of a droplet of volume *iV*_1_, and *ω*(*i*, *j*) is the probability that a fragment formed by the breakup of a droplet of volume *jV*_1_ will have volume *iV*_1_.

Tokita [24] considered the average droplet size in steady shear flow. He assumed that the droplets were still monodisperse in size and derived the following dependence of steady radius of droplet *R* on system parameters:(25)$$R=\frac{12\sigma P_{c}\phi }{\pi \eta_{ap}\dot{\gamma }-4P_{c}\phi E_{DK}}$$
where *η**_αp_* is the apparent viscosity of the blend and *E*_DK_ is the volume energy.

Fortelný and Kovár [24] expanded the function of *F*(*k*) as the Taylor series of *Ca* on the right of *Ca_c_*:(26)$$F\left(k;Ca\right)={\left(\frac{\partial F}{\partial Ca}\right)}_{Ca_{c}}\left(Ca-Ca_{c}\right)+\frac{1}{2}{\left(\frac{\partial^{2}F}{\partial Ca^{2}}\right)}_{Ca_{c}}{\left(Ca-Ca_{c}\right)}^{2}+\ldots$$

Substitution of Equation (26) into Equation (24) leads to the linear dependence of *R* on the volume fraction of droplets *ϕ*:(27)$$R=R_{c}+\frac{4\sigma P_{c}}{\pi \eta_{m}f_{F}}\phi$$

Huneault et al. [18] developed a computational model for droplet size evolution during mixing in a screw extruder. They considered that the blend components showed power-law relations between shear stress and shear rate. They assumed that the droplet deformation took place only within the pressurized screw zones. The authors proposed the following equation for *R*:(28)$$R=R^{0}+{\left(1.5C_{H}Ca_{c}t_{B}^{*}\phi^{8/3}\right)}^{1/2}$$
where *R*^0^ is the droplet radius for *ϕ* = 0, and *t_B_** is the dimensionless breakup time, which is a function of *p* and is independent of *Ca*.

Fortelny and Juza [25] recently formulated equations for the calculations of steady drop size in flowing immiscible polymer blends based on the monodisperse-drop-size assumption.
(29)$$a_{m}{\left(R^{*}-1\right)}^{a_{c}+1/2}+{\left(R^{*}-1\right)}^{1/2}-\frac{4\left(1+p\right)}{\pi }k_{1}P_{c}\left(R\right)\phi \left(R^{*}-1\right)-\frac{4\left(1+p\right)}{\pi }k_{0}P_{c}\left(R\right)\phi =0$$
where *R** is the ratio of *R* and its critical value for breakup, *R**c*, and *a**m* = 8.759 and *a**c* = 1.748, *k*0 = 4.3 and *k*1 = 27.7 are the numerical constants determined from the experimental data of Cristini et al. [26]. The prediction of this model agreed well with the experimental data when the component ratio was low, as can be seen in Figure 4.

For *Ca* >> *Ca_c_*, the approach led to the following equation:(30)$$R^{*1/3}=\frac{4}{\pi }\frac{\phi }{g(p)Ca_{c}^{1/3}}P_{c}(R)$$

Janssen and Meijer [14] studied the evolution of droplet size in an extruder using a two-zone model. The model of a cascade of ideal mixers was used for residence time distribution in the weak zone.

Patlazhan and Lindt [29] solved Equation (24) using an expression for droplet breakup constructed by a combination of the results of Tomotika’s theory [30]. Droplet size distribution function, as a function of the initial droplet size distribution, *p*, and the average number of daughter droplets, was calculated numerically.

Delamare and Vergnes [31] studied the evolution of droplet size distribution in a twin-screw extruder. Average droplet diameters and local distribution of the droplet sizes were calculated numerically as functions of the parameters of blends and the extrusion process.

Potente and Bastian [32] derived an algorithm for the calculation of droplet size evolution during extrusion using the finite and boundary element methods for the determination of stress acts on the droplets during their trajectories.

Peters et al. [33] derived a constitutive equation for liquid mixture based on the Lee and Park model of immiscible polymer blends [34]. A scheme for the calculation of morphology evolution, considering the above events, was proposed. Results of the theory were compared with experimentally determined time dependence of rheological functions in various flow regimes and not with droplet size distribution.

More recently, Wong W.-H. B. et al. [35] extended the constitutive modeling of dispersive mixtures proposed by Peters G.W.M. [33] to study the polydisperse droplet size distribution numerically. The simulation procedure contained an additional morphology state and behaved better in complex flows, i.e., eccentric cylinder flow.

### 3.2. Phase Field Models

It is straightforward to model the polymer blends at continuum length scales in the Cahn–Hilliard–Cook framework [36]. The Cahn–Hilliard equation has multiple complexities from both mathematical and physical viewpoints. Mathematically, as a partial differential equation, it has strong nonlinearity. Physically, the Cahn–Hilliard equation is fundamentally related to the thermodynamics of the polymer blends, and this relationship is reflected in the Gibbs free energy of the mixing function.

In the case of polymer blends, the most common choice is the Cahn–Hilliard model, which is a remarkably simple and powerful equation, so that a wealth of supporting data can be found from the literature.
(31)$$\frac{\partial c}{\partial t}+u\cdot \nabla c=D\nabla^{2}\mu$$
where *c* is the composition of the fluid, ***u*** is the velocity of the flow field, *D* is a diffusion coefficient of the m^2^/s unit, $\mu =c^{3}-c-\gamma \nabla^{2}c$ is the chemical potential at that location, and $\sqrt{\gamma }$ is the thickness of the two-phase transition region (interface).

Later, Cook et al. [37] further modified the Cahn–Hilliard equation by adding a thermal noise term, *ξ*, thus making the equation thermodynamically more complete and presenting the well-known Cahn–Hilliard–Cook equation, namely:(32)$$\frac{\partial c}{\partial t}+\nabla \cdot \left(uc\right)=D\nabla^{2}\mu +\xi$$
where *ξ* satisfies the fluctuation dissipation theorem.
(33)〈ξ(r,t)〉=0
(34)$$\langle \xi \left(r,t\right)\xi \left(r^{\prime },t^{\prime }\right)\rangle =2k_{B}TD\nabla^{2}\delta \left(r-r^{\prime }\right)\delta \left(t^{\prime }-t\right)$$

Although a thermal noise term is added to the equation, the thermal noise only plays an important role in the initial stage of phase separation and has a negligible role in the evolution of the dispersed phase morphology after complete phase separation.

The flux ***j****_i_* of species *i* can be written as
(35)$$j_{i}=-\sum_{j}M_{ij}\nabla \mu_{j}$$

To obtain the transport equation for species *i*, the continuity equation is applied:(36)$$\frac{\partial \phi_{i}}{\partial t}+\nabla \cdot j_{i}=0$$
where *t* denotes time. For an *N* component system, typically, *N* − 1 transport equations are defined, and *ϕ_N_* for the last component is inferred from a material balance equation $\sum_{i}\phi_{i}=1$, where *ϕ_i_* is the volume fraction of species *i*. For a binary system with components 1 and 2, the transport equation for species *i* can then be written as:(37)$$\frac{\partial \phi_{1}}{\partial t}=\nabla \cdot \left(M_{12}\nabla \mu_{12}\right)$$

An expression for the chemical-potential difference, *μ_ij_*, can be obtained by considering the generalized *N*-component Landau–Ginzburg free-energy functional equation for inhomogeneous systems enclosed within a dimensionless volume.
(38)$$\mu_{12}=\frac{\partial g_{m}}{\partial \phi_{1}}-\kappa \nabla^{2}\phi_{1}$$

Prusty and Keestra et al. [38] used a diffusion interface model based on Cahn–Hilliard theory to study the internal structural evolution of poly(methyl methacrylate) (PMMA)/poly(styrene-co-acrylonitrile) (SAN)28 blends and compared it with small-angle scattering (SALS) experiments. The simulation results showed that the coarsening kinetics are mainly dominated by the flow when the capillary number exceeds 10.

Kohler and Krekhov et al. [39], on the other hand, discussed the dynamic critical properties of PDMS/PEMS blends in the framework of the generalized Cahn–Hilliard model and examined the formation process of internal structures in the presence of an inhomogeneous spatial distribution of temperature and found that when the temperature of the spatially periodic temperature field changes by more than a critical value, the process of coarsening will be blocked.

Keestra and Goossens et al. [40] used a diffusion interface model based on Cahn–Hilliard theory to simulate and study the phase morphology of the PMMA/SAN blends under a flat plate shear flow field with increasing shear intensity, and the results obtained were consistent with those of optical microscopy and small-angle light scattering (SALS) experiments.

Parsa and Ghiass et al. [41] developed a kinetic model for the phase separation process of polystyrene/polyvinyl methyl ether blends based on the nonlinear Cahn–Hilliard theory, identified a variety of dispersed phase morphologies such as droplets and rods, and revealed that the initial component concentration of the polymer blends and temperature distribution were the main controlling factors of phase separation and morphology evolution.

Carolan and Chong et al. [42] implemented the Cahn–Hilliard model in the open-source computational fluid dynamics software library OpenFOAM and investigated the “sea-island “ droplets structure and co-continuous structure during processing, revealing that the initial concentration of the two phases has a decisive role in the final phase morphology.

Tabatabaieyazdi and Chan et al. [43] coupled the nonlinear Cahn–Hilliard model with Flory-Huggins-de-Gennes to simulate surface-oriented phase separation phenomena in binary polymer blends and investigated, for the first time, the different temperature gradients on the surface enrichment rate.

In recent years, the development of the Cahn–Hilliard model in immiscible polymer blends showed a trend in integration with the Doi–Ohta model [44,45]. By extracting new coarse-grained variables, microstructural models on different levels were coupled. Through this coarse-grained method, the morphology evolution of polymer blends could be studied from the thermodynamic perspective.

### 3.3. Lattice Boltzmann Method

A major advantage of the Lattice Boltzmann method is its ability to serve as a solver for conservation equations, such as the Navier–Stokes equation describing holonomic flow; at the same time, its mesoscopic nature provides a viable way to incorporate the microscopic dynamics of forming liquid–liquid interfaces and liquid–solid interfaces. In the last two decades, several multiphase models have been developed in the LBM research community. The first is the Rothman–Keller (RK) LB model, proposed by Gunstensen et al. in 1991, based on Rothman and Keller’s lattice gas (cellular-automaton) model [46], which uses a color gradient to achieve phase separation and model the interaction forces at the multiphase interface [47]. The second one is the pseudo-potential model proposed by Shan and Chen [48], which considers nonlocal inter-particle interactions by introducing a pseudo-potential. Another is the free energy model proposed by Swift [49], which proposes a generalized equilibrium state distribution function for a non-ideal pressure tensor, thus introducing multiphase interactions directly into the collision process of the distribution function. Another one is the mean field LB model proposed by He et al. [50] for incompressible multiphases, with suitable external force terms near the interface.

Among the above-mentioned models, the Shan–Chen pseudo-potential multiphase flow model is, to the best of the authors’ knowledge, the most widely used multiphase flow lattice Boltzmann model because it has the advantages of simplicity of approach and generality. Its basic idea is to use the pseudo-potential (also often called effective mass) to reflect the microscopic intermolecular forces on a mesoscopic scale. This automatic phase separation mechanism is an attractive feature of the pseudo-potential model because the two-phase interface is no longer a mathematical boundary and no longer requires any explicit interface tracing or interface capture techniques. Because of its remarkable computational efficiency and clear picture of the underlying microphysics, the Shan–Chen pseudo-potential multiphase model has become a promising technique for simulating and studying polymer blends.

Taking the most widely used single relaxation time LBM-BGK as an example, its distribution function evolution equation is
(39)$$\begin{array}{l} f_{\sigma ,\alpha }\left(x+ce_{\alpha }\Delta t,t+\Delta t\right)-f_{\sigma ,\alpha }\left(x,t\right)\\ =-\frac{1}{\tau_{\sigma ,\alpha }}\left(f_{\sigma ,\alpha }\left(x,t\right)-f_{\sigma ,\alpha }^{eq}\left(x,t\right)\right)+F_{\sigma ,\alpha }\left(x,t\right) \end{array}$$
where *f**_σ_**_,α_*(***x***,*t*) is the probability of finding a fluid particle of component *σ* at position ***x*** and time *t* with the discrete velocity ***e***_α_.

To introduce nonlocal interactions between particles, Shan and Chen defined the force on a particle with component *σ* at spatial position ***x*** from a particle of component $\bar{\sigma }$ at spatial position $x^{\prime }$ as
(40)$$F\left(x,x^{\prime }\right)=-G\left(\left|x-x^{\prime }\right|\right)\psi_{\sigma }\left(x\right)\psi_{\bar{\sigma }}\left(x^{\prime }\right)\left(x^{\prime }-x\right)$$
where *G* is a symmetric Green’s function, and *ψ* is an effective mass that depends on the local density of the components. The inter-particle force separates the different phases automatically, as illustrated in Figure 5.

The force defined in Equation (40), along the vector between the two lattice positions, satisfies Newton’s third law while the global conservation of momentum is maintained [51].

As shown in Figure 5 above, the combined force on the particle at position ***x*** is the sum of the forces acting on it by all neighboring particles.
(41)$$F\left(x\right)=-\psi \left(x\right)\sum G\left(\left|x-x^{\prime }\right|\right)\psi \left(x^{\prime }\right)\left(x^{\prime }-x\right)$$

In the lattice space system, if we consider *N* neighboring particles interacting with the particle at the current position ***x*** and let *G*(|***e***_α_|) be just a function of |***e****_α_*|, i.e., the interaction between the particles is isotropic, then the interaction force ***F***(***x***) can be further expressed as:(42)$$F_{\sigma \bar{\sigma }}\left(x\right)=-g_{\sigma \bar{\sigma }}\psi_{\sigma }\left(x\right)c_{s}^{2}\sum_{\alpha =1}^{N}w\left({\left|e_{\alpha }\right|}^{2}\right)\psi \left(x+e_{\alpha }\right)e_{\alpha }$$
where $g_{\sigma \bar{\sigma }}$ is the interaction strength between component σ and $\bar{\sigma }$ and is linked to the thickness of the interface, and *w*(|***e****_α_*|^2^) is the weight that is used to calculate the isotropic interaction force.

Macroscopic density *ρ*_σ_ and momentum *ρ_σ_**u**_σ_* are defined as the zeroth and first-order moment of the distribution function:(43)$$\begin{array}{l} \rho_{\sigma }=\sum_{i}f_{\sigma ,i}\\ \rho_{\sigma }u_{\sigma }=\sum_{i}e_{i}f_{\sigma ,i} \end{array}$$

In addition, in the Shan–Chen pseudo-potential model, the macroscopic velocity of the flow field as a whole is redefined as the average of the velocities before and after the collision, i.e.:(44)$$\rho u=\sum_{\sigma }\rho_{\sigma }u_{\sigma }+\delta t\sum_{\sigma }F_{\sigma }/2$$

By performing the Taylor expansion of Equation (39) to the second-order by Chapman–Enskog analysis, the continuity equation for each component can be obtained as follows:(45)$$\frac{\partial \rho_{\sigma }}{\partial t}+\nabla \cdot \left(\rho_{\sigma }u\right)=-\nabla \cdot j_{\sigma }$$

Additionally, the continuity equation within the entire flow field is as follows:(46)$$\frac{\partial \rho }{\partial t}+\nabla \cdot \left(\rho u\right)=0$$
where $j_{\sigma }=\rho_{\sigma }\left(u_{\sigma }-u\right)$ is the diffusive mass flux of component *σ*. By the same analysis method, the momentum equation for the entire flow field can also be obtained from Equation (39), when the macroscopic viscosity of the fluid can be written as:(47)$$v_{\sigma }=\frac{1}{\dim+2}\left(\sum_{\sigma }\frac{\rho_{\sigma }}{\rho }\tau_{\sigma }-\frac{1}{2}\right)$$

The above analysis demonstrates that the pseudo-potential mode of LBM is equivalent to the Navier–Stokes equations of multiphase flow with second-order accuracy.

The comparison of the mesoscale methods for droplet morphology evolution is arranged in Table 1 for the future choice of the reader according to different fitness.

## 4. Macroscopic Mold-Filling Flow Simulation

The mold-filing flow of the polymer melt is a non-Newtonian, non-isothermal, and unsteady process with moving free-surface. This complex process is governed by conservation laws and the constitutive equation, and the simulation of mold-filling flow is, in essence, to solve the governing equations numerically.

Mass:(48)$$\frac{\partial \rho }{\partial t}+\rho \nabla \cdot u+u\cdot \nabla \rho =0$$

Momentum:(49)$$\frac{\partial \left(\rho u\right)}{\partial t}+\nabla \cdot \left(\rho uu\right)=-\nabla P+\nabla \cdot \left(\eta \nabla u\right)+\rho f$$

Energy:(50)$$\rho c_{p}\frac{\partial T}{\partial t}+\nabla \cdot \left(\rho Tu\right)=\nabla \cdot \left(\lambda \nabla T\right)+\Phi$$
where ***u***, *P,* and *T* are the velocity, pressure, and temperature of the mold-filling flow; ***f*** is the external force field; *ρ*, *η*, *c_p_*, and *λ* are the density, viscosity, specific heat, and heat conductivity of the polymer melt; Φ is the heat dissipation.

The simulation of the mold-filling flow of polymer melt during injection molding has undergone a tortuous development process from 1D, 2D, 2.5D to 3D. Among them, the research on one-dimensional flow simulation started in the 1960s, and the simulation objects were mainly round tubes with simple geometry or rectangular or centrally cast discs [11,12,13,14,15,16]. Williams et al. [2,17] conducted an exhaustive study of the circular tube flow of plastic melts. 1D flow analysis can obtain the pressure and temperature distribution of the melt. The calculation is fast, and the location of the flow front is easy to determine, but it is limited to simple and regular geometry, which is difficult to adapt to the actual needs of production.

Since plastic injection molded products are generally thin-walled structures, the dimensions in the thickness direction are much smaller than the overall dimensions of the product, and the longer molecular chain structure of the plastic melt leads to strong viscosity, with inertia forces much smaller than the viscous shear stress. Hieber et al. [19,20,21,22] extended the flow model of the Hele–Shaw assumption [52,53] to the two-dimensional flow of polymer melt. The Hele–Shaw model neglects the inertia and gapwise velocity components for polymer melt flow in thin cavities. The flow-governing equations are simplified into a single Poisson equation based on these assumptions.

Simplified Momentum Equation:(51)$$\frac{\partial P}{\partial x}=\frac{\partial }{\partial z}(\eta \frac{\partial v_{x}}{\partial z})$$
(52)$$\frac{\partial P}{\partial y}=\frac{\partial }{\partial z}(\eta \frac{\partial v_{y}}{\partial z})$$
(53)$$\frac{\partial P}{\partial z}=0$$

Simplified Energy Equation:(54)$$\rho C_{P}(\frac{\partial T}{\partial t}+v_{x}\frac{\partial T}{\partial x}+v_{y}\frac{\partial T}{\partial y})=\eta {\dot{\gamma }}^{2}+k\frac{\partial^{2}T}{\partial z^{2}}$$

The mathematical model proposed by Hieber et al. is more in line with the actual situation of plastic injection molding and takes into account the possibility of implementing numerical calculations, so it is followed by many researchers [18,23,24,25].

The 2.5D flow simulation is an approximate 3D simulation method that goes through two stages: a mid-plane model and a surface model. Although both theoretical and application examples have demonstrated that the mid-plane model can accurately simulate the filling flow of polymer melt, it is often difficult to extract the mid-plane from the product, which, in turn, leads to secondary modeling problems in CAE software.

In order to solve the problem of secondary modeling of the mid-surface model, the surface model based on solid technology (surface technology), which preserves all the advantages of the mid-surface model, the two-sided analysis model comes out. The first integrated mathematical model of the surface model was presented by Zhou et al. [14,32,33]. At present, the surface model has already become the mainstream model of injection molding simulation and is widely used in commercial CAE systems [34,35,36,37,38,39,40,41].

With the development of industrial technology, the shape of plastic parts is becoming more and more complex, and wall thickness inhomogeneity is becoming more and more prominent. The limitations of the mid-plane model and the surface model are becoming increasingly obvious. In recent years, the accelerating speed of computers and the continuous progress of numerical analysis technology have laid a solid foundation for the application of three-dimensional modeling and simulation.

Chang [27] et al. presented an implicit finite volume approach to simulate three-dimensional mold filling. This method can more accurately predict the critical three-dimensional phenomena encountered during mold filling than the existing Hele–Shaw analysis model. It has been proven to be a highly effective and flexible tool for simulating mold-filling problems.

Kim [28] et al. used mixed interpolation cells of velocity and pressure and simultaneous interpolation cells to simulate velocity and pressure fields during the mold-filling process, respectively, and concluded that the numerical efficiency of simultaneous interpolation is higher.

Hetu [14] et al. used mini cells to solve the instability problem of velocity and pressure and the Lesaint-Raviart method to eliminate the numerical oscillations caused by convective terms.

Pichelin et al. [29,30] used hybrid cells and the explicit Taylor–Galerkin intermittent finite element method to simulate the mold-filling process, and a fountain effect at the flow front was observed.

Li, Q. et al. simulated melt filling and primary gas penetration in a gas-assisted injection molding (GAIM) process using finite volume and domain extension methods with SIMPLEC technology, and the CLSVOF (coupled level set and volume of fluid) method was employed to capture the moving interfaces [54].

Vietri, U. et al. improved the predictions of the description of pressure profiles by introducing the effect of pressure on viscosity and the effect of cavity deformation during molding [55].

He, L. et al. investigated the three-dimensional (3D) injection molding flow of short fiber-reinforced polymer composites using a smoothed particle hydrodynamics (SPH) simulation method [56].

Liang, J. et al. improved the numerical stability of 3D FVM simulation in plastic injection molding by proposing a novel and robust interpolation scheme for face pressure [57].

Xu, X.Y. et al. enhanced the accuracy, stability, and boundary treatment of the smoothed particle hydrodynamics (SPH) method and investigated the injection molding process of polymer melt based on a generalized Newtonian fluid model [58].

Liu, Q.S. et al. integrated the Rolie–Poly constitutive equation with the continuity, momentum, and level set equations to investigate the role of viscoelasticity of polymer melt during injection mold filling [59].

In summary, the main differences among the three mainstream models of mold-filing flow of polymer melt were compared in several areas, as shown in Table 2. 

## 5. Scale-Bridging Strategies

Since there are two ways to describe fluid flows, namely, the Lagrangian description and the Eulerian description, accordingly, the scale-bridging strategies of multiscale simulation of injection molding of polymer blends can be roughly divided into the particle-based scale-bridging method and parameter-based method, which are used with the macroscale mold-filling flow simulation of the two descriptions separately.

### 5.1. Parameter-Based Methods

For the Eulerian description of fluid dynamics, the physical parameters in the governing equations of the mold-filling flow consist of single-valued conditions. Other than measurements from experiments in conventional ways, these macroscopic physical parameters can be determined from the microscale molecule conformation or mesoscale blend morphology of polymers. With the physical parameters as a bridge, the simulations are coupled on macroscopic and mesoscopic or mesoscopic scales.

Based on the ellipsoidal description of droplets, Yu, W. and C. Zhou [60] proposed a rheological constitutive equation for an immiscible polymer blend by adding the contribution of the interfacial stresses to the Newtonian stresses resulting from components.
(55)$$\tau =2\eta_{m}e^{A}+2\left(\eta_{d}-\eta_{m}\right)\phi e-\frac{\Gamma \phi }{L_{c}}\left(\frac{A}{trA}-\frac{1}{3}\delta \right)$$
where ***A*** is the area tensor, converted from the ellipsoid morphology tensor ***G***.

The explicit expression of Equation (55) is very simple, and it is straightforward to bridge the mesoscale blend morphology and macroscale mold-filling process. However, its drawbacks are just as obvious and have been figured out in the literature [61,62]. First of all, the effect of droplet breakup or coalescence on stress is not included in this model, let alone in the case where the ellipsoidal description fails. What is worse, integrating the viscoelasticity of the polymers into the constitutive model of polymer blends based on the ellipsoid description is tough work. Therefore, it is reasonable to develop the rheological model of polymer blends from the molecular physics of polymers.

The simplest physics model for polymeric fluids is the bead-spring model [63], where a polymer molecule is highly coarse-grained into a pair of beads connected by a spring, as depicted in Figure 6. In spite of its extreme simplicity, the bead-spring model is able to reproduce some complex rheological phenomenon of polymers, such as shearing-thinning [64,65], shear-thickening [66,67], and viscoelasticity [68,69].

However, due to some missing structure details of polymer molecules, it is almost impossible for the bead-spring model to deal with the entanglement of polymer melts, which is inevitable in the processing of polymers.

Molecular models of the dynamics of polymer melts and, more generally, dense polymer solutions began with the famed reptation theory by Gennes [70] and the tube model by Doi and Edwards [71]. Unlike the molecular simulations of small molecule fluids, such as water, the most difficult aspect of the molecular simulations of polymers is the handling of the entanglement problem between molecular chains. Inspired by the notion of reptation and tubes, proposed by Gennes and Doi and Edwards, subsequent research has developed a series of rheological models for polymer melts.

Likhtman, A.E. and R.S. Graham have demonstrated that the molecular tube theory can also provide a route to constructing a family of very simple differential constitutive equations for linear polymers [72], just like the widely-used Giesekus, PTT, Larson, or pom–pom equations.

Hua, C.C. and J.D. Schieber extended the original (mean-field) tube model to realistic two or three-dimensional (3D) multichain situations and proposed the so-called slip-link model [73], where the polymer entanglement was considered as a pair of slip links, together with the repetitive movement of the probe chain along the primitive path. This idea was followed by many succeeding scholars.

For instance, considering that the polymer melt always undergoes entanglement and crystallization, Andreev, M. and G.C. Rutledge recently introduced two new parameters, which are functions of the degree of crystallinity, to modify the slip-link model [74]. The model was validated using experimental datasets for isotactic polypropylene.

Taletskiy implemented the discretization of the slip-link model through a rigorous mathematical derivation and gave its computational algorithm, optimized especially for coarse-grained simulations, clearing the way for future industrial applications [75].

Becerra, D. et al. constructed a hierarchy of strongly connected models for discrete slip-link theory and presented the method of determining the values of the four parameters of the discrete slip-link model, especially the friction parameter from the first principles of simulation [76].

During the past two decades, the slip-link model has achieved noticeable success and progress; however, its limitations cannot be ignored. The slip-link model is largely applicable to linear polymers but fails to capture the complex rheology of star, branched, or crosslinked polymers.

To overcome this drawback, Masubuchi, Y. et al. developed the PCN model [77]. Polymer chains were coarse-grained at a level of entanglement molecular weight, and the entangled network of polymers was represented in a (3D) real space. Using the PCN model, the viscosity of bidisperse polystyrene melts [78], block copolymers [79], star polymers [80], branched polymers [79], and uncrosslinked and crosslinked polymers [81] was studied for both shear and elongational flow fields, in agreement with both the experimental data and the conventional phenomenological constitutive model [82].

After years of development and maintenance, the PCN model was incorporated into the commercial simulator NAPLES and then as a package of the software J-OCTA for multiscale simulation of polymer processing [83].

More recently, Huang, L.H. et al. proposed a multiscale computer simulation scheme to investigate the stress-morphology coupling of a Nylon 6/ACM blend under processing conditions in a real extruder [84]. The composition–stress–morphology relationship of the polymer blends was crucial for processing and was revealed by a large-scale rheological simulator (NAPLES) without recruiting any freely adjustable parameters. Although this research was aimed at the extrusion of polymers, the proposed strategy was universal and is a promising tool for the analysis of injection molding of polymer blends.

On the other hand, deep learning technology is developing rapidly and has brought revolutionary advances in the engineering and science of polymers [85]. Unlike the phenomenological approach, which obtains the parameters of the constitutive model from experiments, or the numerical approach, which obtains the stress response of polymers through simulation, deep learning techniques provide fuzzy prediction on the rheology of polymers through ever-richer datasets and more rational deep learning algorithms.

Tran, H.D. et al. [86] succeeded in predicting dozens of polymer properties, appropriate for a range of applications, using the web-based machine-learning capability of Polymer Genome.

Alqahtani, A.S. [87] developed a novel model to predict the viscosity of HPAM polymers by combining fundamental, physical models and machine learning methods, with a great agreement with the measurement of the ARES G2 rheometer.

To the authors’ knowledge, currently, there is a lack of studies on the rheology of polymer blends using deep learning techniques; the authors suggest that the application of image analysis methods to establish the correlation between the morphology and rheology of polymer blends is a feasible direction.

Although deep learning techniques have recently made many valuable explorations in polymer research, they still face several problems and challenges, such as the need for more datasets to train the models; the underlying algorithms of deep learning still need further optimization, and there is the need to get rid of reliance on existing constitutive models. However, it is undeniable that the use of deep learning techniques for the processing of polymers will be a future trend [88].

### 5.2. Particle-Based Methods

In the mold-filling flow of polymer blends, the immersed droplets also drift with the flow rather than being fixed at certain points. The droplet shape is not only decided by the local flow field but also depends on the history of the flow force exerted on it [2]. The Lagrangian description is more direct and has an inherent advantage over the Eulerian description in coupling macroscopic flow field simulation with the mesoscopic polymer blend morphology simulation.

Since the SPH method was established in 1977 by Lucy [89], Gingold, and Monaghan [90], it has become an alternative to the conventional FDM, FEM, and FVM methods in simulating the injection molding of polymers [3,58,91].

In the SPH method, the polymer blend melt is made of a large number of virtual fluid particles, the motion of which is obtained by integrating Newton’s second law. Take the flow around a column as an example. As shown in Figure 7, each fluid particle contains a given number of droplets that move inside the particles. While the motion of the fluid particles is simulated using the SPH method, the history of the flow field force on the droplets inside the fluid particles is determined, and then the morphology evolution of the polymer blends during the flow is obtained. In turn, the morphology of the fluid particles contributes to the SPH simulation of the particles’ motion.

Because of the mesh-free discreteness characteristics of the SPH method, the SPH method can be coupled with whatever micro- or mesoscopic models of the polymers inside the fluid particles.

Murashima, T. et al. integrated molecular dynamics [92] and the coarse-grained dumbbell model [93] with the SPH method to investigate polymeric flow, taking into account the memory effect of polymers.

Likewise, Sato, T. et al. employed the well-known slip-link model inside fluid particles to study entanglement dynamics at the microscopic level [94].

Lee, J. et al. [95] exploited the fact that both SPH and LBM have good parallel computing properties and utilized the LBM inside the fluid particles to simulate the turbulent flow of polymers, the computational cost of which is usually very high.

Recently, Deng et al. [5] integrated the Eulerian model of mold-filling flow and the Lagrangian droplet trajectory tracking method for injection molding of immiscible polymer blends. As shown in Figure 8, different patterns of morphology evolution of the droplets, along their trajectories, were successfully simulated.

In theory, the particle-based scale-bridging method is more suitable for the simulation of polymeric flow for the unique memory effect of polymers. However, the Lagrangian macroscopic simulation of the mold-filling flow of polymer melt has many drawbacks and is not as mature as the conventional Eulerian methods. Firstly, it is more difficult to prescribe boundary conditions in SPH than other mesh-based methods; secondly, the computational overhead of the SPH method is remarkably larger, reducing the overall efficiency of the multiscale simulation; furthermore, as the mold-filling flow is a convection-dominant process and the SPH method is explicit, the problem of numerical stability must be observed.

## 6. Outlook and Summary

The development of polymer blend products necessitates an in-depth understanding of the processes at different time and length scales during injection molding. This need has greatly promoted the advance in theoretical and numerical methods to model and simulate the inherent hierarchical phenomena in polymer blends. The present review attempts to survey the state-of-the-art of various multiscale simulation approaches as applied to polymer blends.

On the macroscopic scale, simulation of the plastic injection molding filling process is a comprehensive technology based on theories and techniques related to many disciplines, and this involves a wide range of research topics. Although fruitful achievements and progress have been made in academia, there are still some areas needing improvement and refinement:Polymer melts are mostly viscoelastic fluids. Although the generalized Newtonian fluid model has been able to accurately simulate the filling flow process in most cases, for some products with high requirements on mechanical properties, optical properties, or geometric accuracy, the residual stress caused by viscoelasticity during the filling and packing process is often not negligible. Therefore, how to establish a stable and efficient method for solving the viscoelastic flow solution for the actual product forming process is also an important topic worth studying.The energy equation of the filling flow process is significantly convection-dominant, and the boundedness of the discrete scheme of the convection term in the equation has an important impact on the accuracy and stability of the whole filling flow simulation. Therefore, it is necessary to study the discrete scheme of the convection diffusion equation with high accuracy under an unstructured grid to satisfy the boundedness.In the simulation of the mold-filling flow process, the solution of the algebraic equation system occupies most of the computational time, among which the solution of the velocity-pressure coupled algebraic equation system takes the most time. Therefore, for the research and development of efficient solution methods for the velocity-pressure-coupled algebraic equation system, shortening the process is also an important part of the next work.

On the mesoscopic scale, some preliminary work has been done for the morphology evolution of individual droplets in simple shear or tensile flow fields, but much remains to be explored and studied with respect to the microstructure evolution inside the blends in more complex injection molding processes:When the fraction of the blends exceeds a certain range (greater than about 40%), the dispersed phase no longer exists in the form of isolated droplets, and the simulation algorithm based on the ellipsoidal assumption becomes invalid and other morphology models could be considered, such as the interfacial tensor model.Most current models of droplet morphology evolution are limited to Newtonian fluids due to the non-uniformity and strong nonlinearity of the viscoelastic constitutive equations. However, the elasticity of the polymer melt has a significant effect on the evolution of droplet morphology, so the role of component elasticity on phase morphology should be considered, for example, by introducing empirical parameters into the models.The evolution equation of droplet size distribution is an important way to parameterize the microstructure of the blend. However, the current evolution models are still based on the ellipsoidal droplet assumption, which cannot characterize the complex morphology of the dispersed phase, so there is a need to establish the evolution equations of the dispersed phase distribution based on the tensor form or the component concentration form in the future.

In terms of macroscopic and mesoscopic scale bridging, particle-based scale coupling methods have been increasingly focused on and applied due to the unsteady flow of the injection molding process and the complex molecular structure of polymers, but there are still some key issues to be studied and solved:Compared with traditional Eulerian methods, such as the finite volume method and the finite element method, the SPH method, based on the Lagrangian description, has the natural advantage of automatically recording polymer history in simulating polymer melt flow; however, poor numerical stability, high computational cost, and difficulty in boundary handling confine its further application in simulating polymer processing, which needs to be addressed in the future.The rheological constitutive relationship of polymers is the key to realizing the coupling between macroscopic and mesoscopic scales; however, it is not easy to establish the constitutive relationship of polymer blends in traditional equation form. It is a promising alternative to use the current data-driven modeling method based on deep learning to propose the constitutive relationship of polymer blends.Since the size of the dispersed phase droplets in the blend is very small and their number is very large, simulating the morphological evolution of the entire dispersed phase during the mold-filling flow is still unaffordable under current computing power, so it is necessary to investigate the use of parallel computing and GPU accelerometers to increase the efficiency of the simulation and the use of multidimensional fractal theory for the parametric description of dispersed phase morphology.

For injection molding of polymer blends, developing multiscale modeling and simulation methods could lead to the design of products simultaneously, on many scales, instead of trial-and-error experimentations. Although it will not be easy in the forthcoming years, it undeniably represents the remarkable value of the future of polymer science.

## Figures and Tables

**Figure 1 polymers-13-03783-f001:**
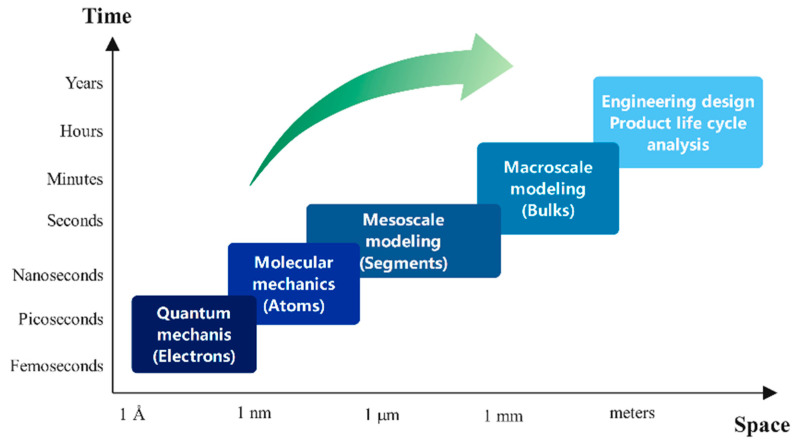
Spatial and temporal multiscale of polymer.

**Figure 2 polymers-13-03783-f002:**
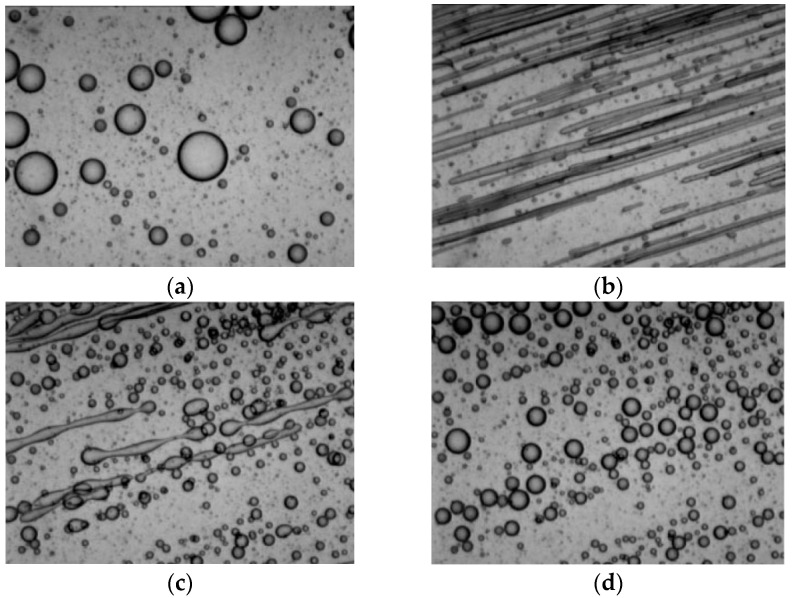
Morphology of the dispersed phase in PDMS/PB blends. (**a**) Before shearing; (**b**) during shearing; (**c**) 25 s after shearing stop; (**d**) 89 s after shearing stop (experimental results from the literature [6]).

**Figure 3 polymers-13-03783-f003:**
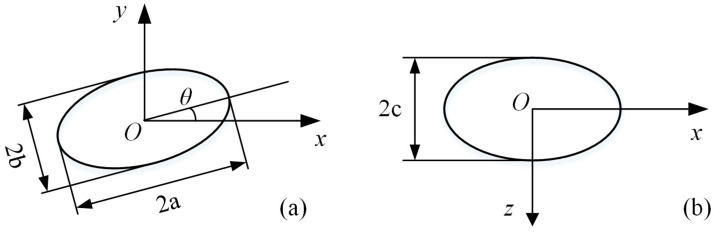
Schematic diagram of droplet geometry model parameters in the (**a**) flow plane and (**b**) rotational plane.

**Figure 4 polymers-13-03783-f004:**
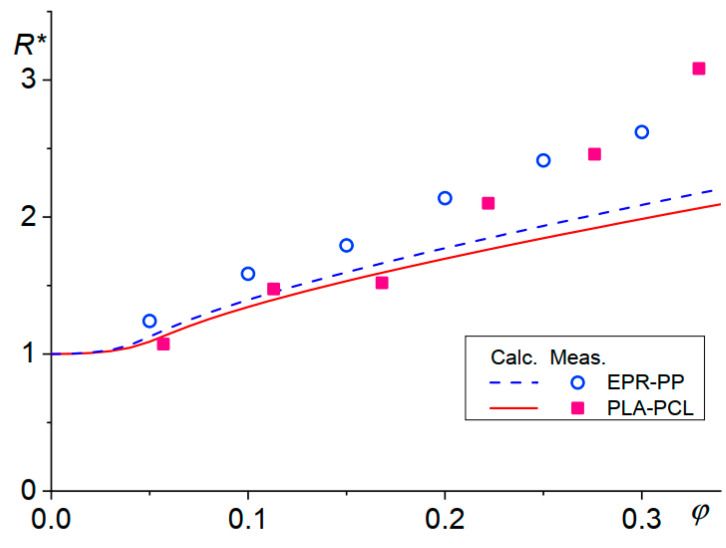
Reduced droplet sizes calculated using Equation (24) and the experimental sizes for blends ethylene-propylene rubber (EPR)-polypropylene (PP) [27] (dashed line and empty circles) and poly(lactic acid) (PLA)-poly(caprolactone) (PCL) [28] (solid line, full squares).

**Figure 5 polymers-13-03783-f005:**
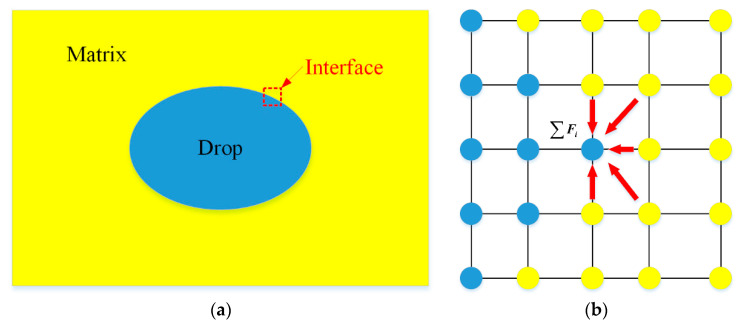
The principle of the pseudo-potential model: (**a**) the schematic of the drop–interface–matrix structure; (**b**) the pseudo-potential force between particles at the interface.

**Figure 6 polymers-13-03783-f006:**
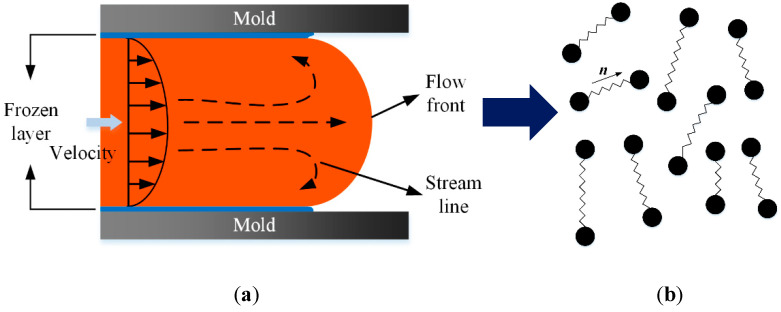
Procedure of the parameter-based scale-bridging methods: (**a**) macroscopic mold-filling simulation; (**b**) polymer chains conformation.

**Figure 7 polymers-13-03783-f007:**
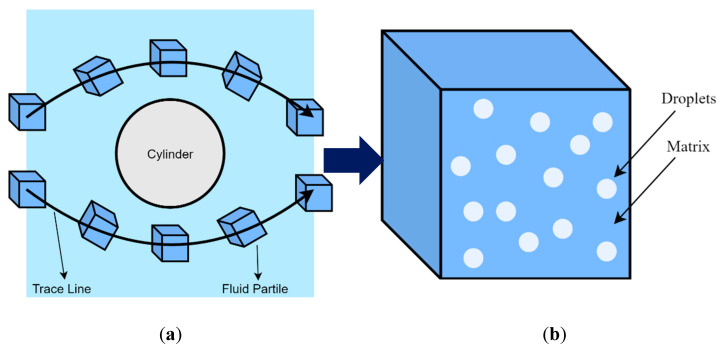
Particle-based scale-bridging method: (**a**) a fluid particle trace in the flow around circular cylinder; (**b**) a fluid particle containing droplets.

**Figure 8 polymers-13-03783-f008:**
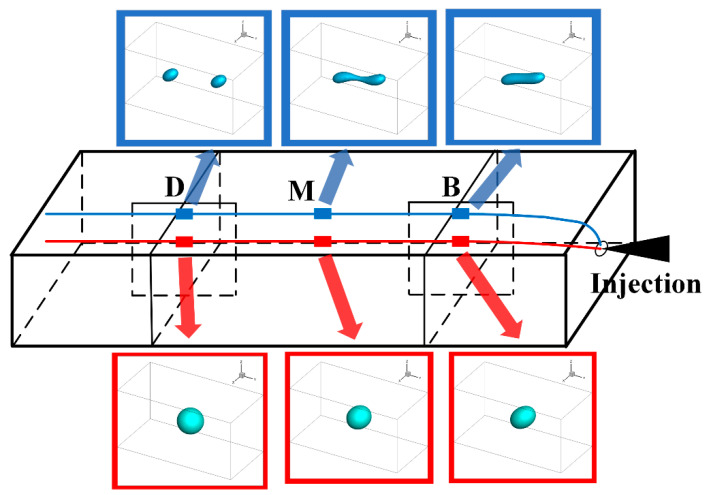
Droplet morphology evolution along their trajectories during mold-filling.

**Table 1 polymers-13-03783-t001:** Comparison of three droplet morphology evolution models.

	Ellipsoid Models	Phase Field Models	LBM
Morphology	ellipsoid	arbitrary	arbitrary
Interface tracking	√	×	×
Interface type	sharp	diffuse	diffuse
Flow filed inside drops	×	√	√
Physical domain size	large	medium	small
Source of model parameters	physical properties	first-principles calculations	physical properties
Solving method	implicit	implicit	explicit
Computation cost	low	medium	high
External field incorporation	×	√	√
Phase transition incorporation	×	√	√

**Table 2 polymers-13-03783-t002:** Differences among three models.

	Mid-Plane Model	Surface Model	Solid Model
Thin-wall laminar flow assumptions	√	√	×
Incompressibility assumption	√	√	√
Inertia and volume forces	×	×	×
Heat transfer in direction of flow	√	√	×
Internal heat source items	×	×	×
Constant physical parameters	√	√	√
Planar-shaped flow front	√	√	×
Grid size	small	medium	large
Algorithm complexity	simple	complex	more complex
Calculation time	short	ordinary	long

## Data Availability

The data presented in this study are openly available in corresponding references.

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
