# Peer review of "Multiscale Modeling and Simulation of Polymer Blends in Injection Molding: A Review"

_polymers, 2021, doi:10.3390/polym13213783_

Round 1

Reviewer 1 Report

While I appreciate the efforts of the authors to provide a review of polymer blends in flow from a modelling and simulation point of view, the manuscript in its present form is in my view far from acceptable for publication. 

For a start, I find it almost disrespectful to submit a manuscript with so many formal issues regarding the presentation of the material. Below is an incomplete list, collected until I run out of patience. 

Second, the paper appears as a rather loose connection of different pieces, whereas the benefit of a good review would in my view be to provide the big picture and relate different approaches to each other. 

Third, the multiscale aspect is discussed rather poorly in Sect 2 and boils down to the CONNFFESSIT idea proposed in Ref [66].
In Sect. 5, more recent approaches are mentioned. But again, the presentation resembles a list of approaches rather than a focused review. Since there are way too many works on scale-bridging in polymers to be listed here, it would be more appropriate to focus on those relevant for polymer blends and/or flow and point our interrelations, benefits and drawbacks as well a potential for future work. 

Fourth, for most of the presentation, I do not see an immediate connection to injection molding. Would be helpful for readers to better keep focus.  

Below are some more detailed comments: 

1. In a review, I would expect that Eq (2) is at least briefly discussed or assumptions mentioned. In particular that the relevant interfaces are sufficiently smooth such that they can be described by a second order anisotropy tensor. This implies that droplet should be of ellipsoidal shape. 

2. Discussion of Fig 3 is not satisfactory, since the multiscale aspect, in particular the mesoscale model is not discussed. A closed constitutive model such as Eq 3 with specified parameters would not need an additional mesoscale model. This lack of discussion is even more annoying since this paper is meant as a review on multiscale modelling and simulation. 

3. Sect 3.1.3 on breakup: I think it would be appropriate to comment that the basic equation (3) for the stress is no longer applicable. How does one proceed to calculate stresses then? 

4. Notation on the breakup frequency is unclear to me. In Eq (25), F(k) is introduced as overall breakup frequency, but seems to refer to the number of droplets of volume k? In Eq 27, F has lost its argument, in (26) it reads Fn where n is the number of droplets, while in (29) the argument is Ca. 
Please use consistent notations throughout! 

5. The Doi-Ohta model [M. Doi and T. Ohta, J. Chem. Phys. 95, 1242 1991] is a classic for the non-equilibrium dynamics of polymer blends. Why is it not even mentioned here, including scale-bridging works associated to it? 

Further:

Line 50: "rotation radius" -> gyration radius

Line 51: Why would asphalt be classified as polymer blend? A more suitable example seems in order. 

Line 53, 88, 134: reference for figure missing. Also at several places later. 

Line 56: why would quantum effects matter? Time scales of 10^-15 s are not seriously an issue even for atomistic simulations. 

Line 105, 107, 108: D and sigma should appear as boldface math symbols. 

Line 115: Dm and Dd should be set in math boldface. 

Line 115: Does Dd correspond to the dispersed phase? Then the order should be reversed. 

Line 117: reference 98 does not exist. 

Line 123/124: equation should be set in math font with correct subscripts. 

The quantity p (viscosity ratio, I guess) is not defined in Equation (3). 

Figures 2 and 3 are almost identical. 

The relation between the anisotropic tensor q used in Eq (2) and (3) and the parameters D and theta (line 173) should be pointed out. 

Why is a new tensor G introduced in Eq (4) in addition to q? This seems unnecessary and not helping the reader.

Shouldn't the tensor L in Eq (4) be identical to Dd? If it is, do not introduce another symbol. If it is not, please explain the difference.  

Line 206: "MM model [6]." is not a proper sentence. Neither are the corresponding ones in line 215, 221.

Line 207: The symbol tau was already used as stress in Eq (2). 

Line 212: "the asymptotic method" is unclear. In which limit is the model required to reproduce which results? 

Line 173: Maybe use a different symbol since D already used as deformation tensor. 

Line 239, 244, 254: "Cacrit" should be set in math mode with proper subscript. 

How does the stress tensor tau in Eq (52) relate to tau_m and tau_d in Eq (2)? 

Author Response

Dear Editors and Reviewers:

Thank you for your letter and for the reviewers’ comments concerning our manuscript entitled “Multiscale Modeling and Simulation of Polymer Blends in Injection Molding: A Review” (ID: polymers-1377226). Those comments are all valuable and very helpful for revising and improving our paper, as well as the important guiding significance to our researches. We have studied comments carefully and have made correction which we hope meet with approval. Revised portion are marked in red in the paper. The main corrections in the paper and the responds to the reviewer’s comments are as flowing:

Reviewer 1:

“While I appreciate the efforts of the authors to provide a review of polymer blends in flow from a modelling and simulation point of view, the manuscript in its present form is in my view far from acceptable for publication.

“For a start, I find it almost disrespectful to submit a manuscript with so many formal issues regarding the presentation of the material. Below is an incomplete list, collected until I run out of patience.

“Second, the paper appears as a rather loose connection of different pieces, whereas the benefit of a good review would in my view be to provide the big picture and relate different approaches to each other.

Reply: Section was rewritten and stressed the connection between the sections of this paper and took the advice of the reviewer to strengthen the ties between scale of macro and meso scales.

“Third, the multiscale aspect is discussed rather poorly in Sect 2 and boils down to the CONNFFESSIT idea proposed in Ref [66].

Reply: After the reviewer’s comments, the author considered this presentation of multiscale framework is only one type and not general, so dropped this.

“In Sect. 5, more recent approaches are mentioned. But again, the presentation resembles a list of approaches rather than a focused review. Since there are way too many works on scale-bridging in polymers to be listed here, it would be more appropriate to focus on those relevant for polymer blends and/or flow and point our interrelations, benefits and drawbacks as well a potential for future work.

Reply: Thanks a lot for this comment. The author rewrote Section 5 and follow the development of major molecular models to reorganize the material, highlighting the importance of rheology in terms of scale coupling. The advantages, disadvantages, difference and connection between various scale coupling methods were added.

“Fourth, for most of the presentation, I do not see an immediate connection to injection molding. Would be helpful for readers to better keep focus.

Reply: In the revised manuscript, the author reselected reference papers and confined the scope within papers concerning polymer processing, i.e., extruding, injection molding.

“Below are some more detailed comments:

“1. In a review, I would expect that Eq (2) is at least briefly discussed or assumptions mentioned. In particular that the relevant interfaces are sufficiently smooth such that they can be described by a second order anisotropy tensor. This implies that droplet should be of ellipsoidal shape.

Reply: Eq (2) of the original paper has been relocated in Section 5, where is more proper. The prerequisites for the equation to hold was also stated.

“2. Discussion of Fig 3 is not satisfactory, since the multiscale aspect, in particular the mesoscale model is not discussed. A closed constitutive model such as Eq 3 with specified parameters would not need an additional mesoscale model. This lack of discussion is even more annoying since this paper is meant as a review on multiscale modelling and simulation.

Reply:

“3. Sect 3.1.3 on breakup: I think it would be appropriate to comment that the basic equation (3) for the stress is no longer applicable. How does one proceed to calculate stresses then?

“4. Notation on the breakup frequency is unclear to me. In Eq (25), F(k) is introduced as overall breakup frequency, but seems to refer to the number of droplets of volume k? In Eq 27, F has lost its argument, in (26) it reads Fn where n is the number of droplets, while in (29) the argument is Ca.

Please use consistent notations throughout!

“5. The Doi-Ohta model [M. Doi and T. Ohta, J. Chem. Phys. 95, 1242 1991] is a classic for the non-equilibrium dynamics of polymer blends. Why is it not even mentioned here, including scale-bridging works associated to it?

“Further:

“Line 50: "rotation radius" -> gyration radius

Reply: the word was corrected

“Line 51: Why would asphalt be classified as polymer blend? A more suitable example seems in order.

Reply: A more suitable example was used.

“Line 53, 88, 134: reference for figure missing. Also at several places later.

“Line 56: why would quantum effects matter? Time scales of 10^-15 s are not seriously an issue even for atomistic simulations.

“Line 105, 107, 108: D and sigma should appear as boldface math symbols.

“Line 115: Dm and Dd should be set in math boldface.

“Line 115: Does Dd correspond to the dispersed phase? Then the order should be reversed.

“Line 117: reference 98 does not exist.

“Line 123/124: equation should be set in math font with correct subscripts.

“The quantity p (viscosity ratio, I guess) is not defined in Equation (3).

“Figures 2 and 3 are almost identical.

“The relation between the anisotropic tensor q used in Eq (2) and (3) and the parameters D and theta (line 173) should be pointed out.

“Why is a new tensor G introduced in Eq (4) in addition to q? This seems unnecessary and not helping the reader.

“Shouldn't the tensor L in Eq (4) be identical to Dd? If it is, do not introduce another symbol. If it is not, please explain the difference. 

“Line 206: "MM model [6]." is not a proper sentence. Neither are the corresponding ones in line 215, 221.

“Line 207: The symbol tau was already used as stress in Eq (2).

“Line 212: "the asymptotic method" is unclear. In which limit is the model required to reproduce which results?

“Line 173: Maybe use a different symbol since D already used as deformation tensor.

“Line 239, 244, 254: "Cacrit" should be set in math mode with proper subscript.

“How does the stress tensor tau in Eq (52) relate to tau_m and tau_d in Eq (2)?”

Reply to above: the expression and writing mistakes listed above has been corrected.

Firstly, the author feels very astonished and deeply apologize for making that many formal issues regarding the presentation of the material. The authors are quite grateful to the reviewers for patiently and carefully pointing out the various errors in the article. In addition to the issues pointed out by the reviewers, the authors closely checked the entire manuscript and corrected each of these errors.

Moreover, the reviewer’s comments on “Section 2 Framework of Multiscale Modeling” and “Section 5 Scale-Bridging Strategies” have inspired the authors to rethink the coupling of macroscale and mesoscale models and strengthen its link with polymer processing, and then reorganized and rewrote these two sections. At the same time, the authors modified some confusing symbols from the readers' perspective to make the paper more readable. These comments of the reviewers are of substantial benefit to improve the academics and quality of this paper, and again, the authors sincerely thank the reviewers for the warm help!

Reviewer 2 Report

The paper "Multiscale Modeling and Simulation of Polymer Blends in Injection Molding: A Review" considers the main literature concerning the polymer blend injection molding simulation. The simulation of injection molding of polymer blends is challenging due to different phenomena interacting during the process. The paper is generally well written, however, there are some issues that the authors have to be addressed.

Figures in the main text are not cited correctly.

Pay attention to the symbols used in the equations, sometimes the ones adopted in the equations are different from those adopted in the text. Every symbol adopted in the equations has to be defined in the text.

page 3 "General coupling method" the approach mentioned in this part of the paper was proposed and adopted in the literature. I suggest adding some reference [10.1016/j.polymer.2021.123850; 10.1016/j.mechmat.2019.103225].

page 4 line 123 the equation is not clear

pag 4 Fully coupled method. The accounting of interfacial tension is not common in the simulation of the injection molding process. The author should mention the papers in which this approach was proposed [10.3390/polym13010133] and the modification of the model for the accounting of these effects.

equation 4 please define k, j, and i. Please use the appropriate notation for the equation (tensors and vectors in bold)

pag 6 line 206 define the acronym before using it (MM model??)

pag 7 line 215 define the acronym before using it (JT model)

pag 7 line 224 The authors must define each symbol adopted in the equation, at least using a table at the beginning of the manuscript.

pag 7 line 234 is gamma the shear rate?

equation 34 the authors used symbol D also for one of the droplet dimensions, it would be better to use a different symbol for the diffusion coefficient.

line 433 please define the acronyms

equation 52 pay attention!! [Bird Transport phenomena 2002 pag 84]

equation 53 pay attention!! [Bird Transport phenomena 2002 pag 337]

line 562 the Hele-shaw approach was adopted in many fundamental works on injection molding simulation [10.3139/217.3249; 10.1149/2.0291909jes].

Figure 9 there is some problem with this figure

How the approaches mentioned in paragraph 5 are related to polymer blend processing? The author should discuss this relationship.

Author Response

Dear Editors and Reviewers:

Thank you for your letter and for the reviewers’ comments concerning our manuscript entitled “Multiscale Modeling and Simulation of Polymer Blends in Injection Molding: A Review” (ID: polymers-1377226). Those comments are all valuable and very helpful for revising and improving our paper, as well as the important guiding significance to our researches. We have studied comments carefully and have made correction which we hope meet with approval. Revised portion are marked in red in the paper. The main corrections in the paper and the responds to the reviewer’s comments are as flowing:

Reviewer 2:

“The paper "Multiscale Modeling and Simulation of Polymer Blends in Injection Molding: A Review" considers the main literature concerning the polymer blend injection molding simulation. The simulation of injection molding of polymer blends is challenging due to different phenomena interacting during the process. The paper is generally well written, however, there are some issues that the authors have to be addressed.

“Figures in the main text are not cited correctly.

Reply: The paper was checked and it’s ensured all figures and tables were cited correctly.

“Pay attention to the symbols used in the equations, sometimes the ones adopted in the equations are different from those adopted in the text. Every symbol adopted in the equations has to be defined in the text.

Reply: The paper was checked and it’s ensured all symbols were defined correctly.

“page 3 "General coupling method" the approach mentioned in this part of the paper was proposed and adopted in the literature. I suggest adding some reference [10.1016/j.polymer.2021.123850; 10.1016/j.mechmat.2019.103225].

Reply: Thanks for the supplemental advice of the reviewer! The author rewrote Section II thus there was a little change to the reference.

“page 4 line 123 the equation is not clear

Reply: The inline equation was corrected.

“page 4 Fully coupled method. The accounting of interfacial tension is not common in the simulation of the injection molding process. The author should mention the papers in which this approach was proposed [10.3390/polym13010133] and the modification of the model for the accounting of these effects.

Reply: Thanks for this advice. The method of accounting of interfacial tension was referred in Section 2.

“equation 4 please define k, j, and i. Please use the appropriate notation for the equation (tensors and vectors in bold)

Reply: The lower corner marker was defined.

“pag 6 line 206 define the acronym before using it (MM model??)

Reply: The full name of MM model was presented when it firstly appeared.

“pag 7 line 215 define the acronym before using it (JT model)

Reply: The full name of JT model was presented when it firstly appeared.

“pag 7 line 224 The authors must define each symbol adopted in the equation, at least using a table at the beginning of the manuscript.

Reply: Thanks for this advice very much! The author listed all the symbols of this paper in Appendix at the end of this paper.

“pag 7 line 234 is gamma the shear rate?

Reply: Yes, gamma is the shear rate and corrected.

“equation 34 the authors used symbol D also for one of the droplet dimensions, it would be better to use a different symbol for the diffusion coefficient.

Reply: To avoid confusion, the author use Df instead of D to indicate the deformability of the droplet.

“line 433 please define the acronyms

Reply: It’s defined in the revised manuscript.

“equation 52 pay attention!! [Bird Transport phenomena 2002 pag 84]

Reply: The governing equations were corrected in standard form.

“equation 53 pay attention!! [Bird Transport phenomena 2002 pag 337]

Reply: The governing equations were corrected in standard form.

“line 562 the Hele-shaw approach was adopted in many fundamental works on injection molding simulation [10.3139/217.3249; 10.1149/2.0291909jes].

Reply: Thanks! The important references were added.

“Figure 9 there is some problem with this figure

Reply: The duplicate Fig.9 was removed.

“How the approaches mentioned in paragraph 5 are related to polymer blend processing? The author should discuss this relationship.”

Reply: In Section 2, several typical examples using the approaches on various scales were added which were related to polymer blend processing so as to make the description more vivid.

Firstly, the author feels very astonished and deeply apologize for making that many formal issues regarding the presentation of the material. The authors are quite grateful to the reviewers for patiently and carefully pointing out the various errors in the article. In addition to the issues pointed out by the reviewers, the authors closely checked the entire manuscript and corrected each of these errors.

Moreover, as a research review, to make clear of numerous symbols appearing in the paper is always annoying, so the reviewer’s suggesting of a symbol table is of great meaning. Therefore, the authors supplement a symbol table in the Appendix at the end of this paper. At the same time, the authors modified some confusing symbols from the readers' perspective to make the paper more readable.

These comments of the reviewers are of substantial benefit to improve the academics and quality of this paper, and again, the authors sincerely thank the reviewers for the warm help!

Round 2

Reviewer 1 Report

The authors have made serious improvements to the manuscript which I believe has improved the quality significantly. 
Below are a number of comments on the revised version.

In the discussion of the Doi-Ohta model, it is important to note that (i) there is an additional scalar configurational variable Q, describing the interfacial area per unit volume, and (ii) that - due to the underlying closure problem - different expressions for the convective terms have been proposed, leading to convective nonlinearities. 

Before Eq (3): the formulation "their values are unspecified.." seems odd. Maybe state something like that these are phenomenological model parameters for which certain choices have been proposed. 

On page 6, in the JT and Yu model, is the quantity R the equivalent radius introduced in 3.1.1? If yes, please briefly mention this explicitly, otherwise use a different symbol. 

Eq 6: the tensor R is not defined. 

Eq (29): First, the convective term is usually given by u.(nabla c), rather than nabla.(u c). 
Second, this equation is usually not referred to as Flory-Huggins but as the Cahn-Hilliard model. 
Finally, the interpretation of c is normally the composition, not concentration, since the model describes phase separation. 

Eq (32) I don't think there should be a minus sign for the covariance. 

Interpretation of c in Eq (29) in terms of composition also allows to better connect to multi-component systems, Eqs (33-36). 

I appreciate table I as a nice overview. A comment on "input data access" seems in order as this does not appear self-explanatory to me. 

Eq (53): There has been a large debate about constitutive equations for such cases. An additional reference summarising the debate would be helpful. 

Two lines after Eq (53): "Eq (55) is very simple .." - Eq (55) does not exist in this manuscript. 

End of the same paragraph: "resealable" -> reasonable? But is it really reasonable to try to build this from "atomic" polymer physics? Maybe you mean "molecular"?

I think it is worth mentioning here that efforts to connect Cahn-Hilliard and Doi-Ohta models have been undertaken, see e.g. Jelic, Ilg, Ottinger, Phys Rev E 2010. 

Author Response

Dear Editors and Reviewers:

Thank you for your letter and for the reviewers’ comments concerning our manuscript entitled “Multiscale Modeling and Simulation of Polymer Blends in Injection Molding: A Review” (ID: polymers-1377226). Those comments are all valuable and very helpful for revising and improving our paper, as well as the important guiding significance to our research. We have studied comments carefully and have made correction which we hope meet with approval. Revised portion are marked in red in the paper. The main corrections in the paper and the responds to the reviewer’s comments are as flowing:

Reviewer 1:

The authors have made serious improvements to the manuscript which I believe has improved the quality significantly.

Below are a number of comments on the revised version.

In the discussion of the Doi-Ohta model, it is important to note that (i) there is an additional scalar configurational variable Q, describing the interfacial area per unit volume, and (ii) that - due to the underlying closure problem - different expressions for the convective terms have been proposed, leading to convective nonlinearities.

Reply: Thanks for the reminder. The noteworthy details has been supplemented in the discussion of the Doi-Ohta model.

Before Eq (3): the formulation "their values are unspecified.." seems odd. Maybe state something like that these are phenomenological model parameters for which certain choices have been proposed.

Reply: Thanks for this comment. About determination of the model parameters, the author added explanation to clarify.

On page 6, in the JT and Yu model, is the quantity R the equivalent radius introduced in 3.1.1? If yes, please briefly mention this explicitly, otherwise use a different symbol.

Reply: Thanks for this reminder. In fact, in the two models, R represent different meaning. To avoid confusing, the author modified the symbols in order to make it more consistent.

Eq 6: the tensor R is not defined.

Reply: As one of the manners to avoid confusion, the symbol R was changed to be r and defined.

Eq (29): First, the convective term is usually given by u.(nabla c), rather than nabla.(u c). Second, this equation is usually not referred to as Flory-Huggins but as the Cahn-Hilliard model. Finally, the interpretation of c is normally the composition, not concentration, since the model describes phase separation.

Reply: Thanks for such detailed comment. The author has corrected relevant content of the convective term in the equation and the definition of c.

Eq (32) I don't think there should be a minus sign for the covariance.

Reply: After repeated check, I found it was an error and remove the minus sign.

Interpretation of c in Eq (29) in terms of composition also allows to better connect to multi-component systems, Eqs (33-36).

Reply: The author quite agrees. The notion issue should be taken rigorously.

I appreciate table I as a nice overview. A comment on "input data access" seems in order as this does not appear self-explanatory to me.

Reply: By using the term "input data access", the author means how to acquire the model parameters, via direct experiment measurements, empirical setting or first principal calculation, and whether it’s convenient to acquire them. To make it easy to understand, the author changed the expression to “source of model parameters”.

Eq (53): There has been a large debate about constitutive equations for such cases. An additional reference summarizing the debate would be helpful.

Reply: Thanks for the advice. To enhance the debate more convincing, several reference has been added.

Two lines after Eq (53): "Eq (55) is very simple .." - Eq (55) does not exist in this manuscript.

Reply: The error has been corrected.

End of the same paragraph: "resealable" -> 1? But is it really reasonable to try to build this from "atomic" polymer physics? Maybe you mean "molecular"?

Reply: I am sorry that both are errors and corrected accordingly.

I think it is worth mentioning here that efforts to connect Cahn-Hilliard and Doi-Ohta models have been undertaken, see e.g. Jelic, Ilg, Ottinger, Phys Rev E 2010.

Reply: Thanks for the supplement. After retrieving the literature, the author thought it’s an interesting idea to combine the Doi-Ohta and Cahn-Hilliard models on different levels to develop a coarse-grained method. So the author added relevant references at the end of the discussion of the Cahn-Hilliard model.

Submission Date

27 August 2021

Date of this review

15 Oct 2021 16:54:22

Reviewer 2 Report

The work can be published in the present form.

Author Response

Dear Editors and Reviewers:

Thank you for your letter and for the reviewers’ comments concerning our manuscript entitled “Multiscale Modeling and Simulation of Polymer Blends in Injection Molding: A Review” (ID: polymers-1377226). Those comments are all valuable and very helpful for revising and improving our paper, as well as the important guiding significance to our research. We have studied comments carefully and have made correction which we hope meet with approval. Revised portion are marked in red in the paper. The main corrections in the paper and the responds to the reviewer’s comments are as flowing:

Reviewer 2:

Comments and Suggestions for Authors

The work can be published in the present form.

Reply: Thanks! The author sincerely appreciated the approval and help from the reviewer to improve this review!

Submission Date

27 August 2021

Date of this review

10 Oct 2021 16:31:21
